# Noninvasive Prenatal Screening for Trisomy 21 in Patients with a Vanishing Twin

**DOI:** 10.3390/genes13112027

**Published:** 2022-11-03

**Authors:** Pascale Kleinfinger, Armelle Luscan, Léa Descourvieres, Daniela Buzas, Aicha Boughalem, Stéphane Serero, Mylène Valduga, Detlef Trost, Jean-Marc Costa, Alexandre J. Vivanti, Laurence Lohmann

**Affiliations:** 1Laboratory CERBA, 7/11 Rue de l’Équerre, 95310 Saint-Ouen-l’Aumône, France; 2Obstetrics Center, Jeanne de Flandre Hospital, CHRU Lille, 59000 Lille, France; 3Department of Obstetrics and Gynecology, René Dubos Hospital, 95300 Pontoise, France; 4Department of Obstetrics and Gynecology, DMU Santé des Femmes et des Nouveau-nés, Antoine Béclère Hospital, AP-HP, Université Paris Saclay, 92140 Clamart, France

**Keywords:** noninvasive prenatal testing, vanishing twins, trisomy 21, trisomy 18, trisomy 13, positive predictive value

## Abstract

A vanishing twin (VT) occurs in up to 30% of early diagnosed twin pregnancies and is associated with an increased risk of fetal aneuploidy. Here, we describe our experience in a large VT population of 847 patients that underwent noninvasive prenatal testing (NIPT) for common fetal trisomies over a three-year period. All patients underwent an ultrasound examination prior to NIPT. Two comparison populations were included, namely, the singleton (*n* = 105,560) and the viable multiple gestation pregnancy samples (*n* = 9691) collected over the same period. All NIPT samples in the VT population received a result, of which 14 were high-risk for trisomy 21 (1.6%), nine for trisomy 18 (1.1%), and six for trisomy 13 (0.7%). Diagnostic testing confirmed the presence of trisomy 21 in 6/12 samples, giving a positive predictive value of 50%. One trisomy 18 case and no trisomy 13 cases were confirmed. The time between fetal demise and NIPT sampling did not appear to affect the number of true- or false-positive cases. In conclusion, NIPT is an effective screening method for trisomy 21 in the surviving fetus(es) in VT pregnancies. For trisomies 18 and 13, a positive NIPT should be interpreted carefully and ultrasound monitoring is preferrable over invasive diagnostic testing.

## 1. Introduction

A vanishing twin (VT), first described in 1976, is the disappearance of an embryo and/or gestational sac following documented fetal cardiac activity in both fetuses of a twin gestation [1,2]. The incidence of VT has been shown to range from 10% to 39% in IVF (in vitro fertilization) pregnancies [3,4,5,6,7,8,9,10], whilst the prevalence of VT following spontaneous conception is still unclear [11,12,13]. Vanishing twins have been reported to be associated with adverse perinatal outcomes as well as an increased rate of fetal aneuploidy [13,14].

Prenatal screening in patients with a VT pregnancy can prove difficult. For maternal serum screening (MSS), the presence of a VT can cause significantly elevated levels of the maternal serum marker PAPP-A, which could result in affected pregnancies not being detected [15]. The presence of a VT can also impact noninvasive prenatal testing (NIPT), which screens for fetal aneuploidy using cell-free (cf) DNA in maternal plasma that is released from placental trophoblastic cells. Studies have shown that cfDNA from the vanishing twin can still be present in the maternal plasma up to at least 8 weeks post fetal demise [14], and possibly as long as 15 weeks [16]. Because it is well-known that chromosomal anomalies are a main cause of miscarriages, trisomies could be the cause of the VT, and thus may result in a high percentage of false-positive calls. Recent guidelines from the American College of Obstetricians and Gynecologists and the Society of Maternal-Fetal Medicine recommend that diagnostic testing is offered in multifetal gestations if a VT is identified due to the significant risk of an inaccurate result if serum-based aneuploidy screening or cfDNA screening is used [17]. However, because of an iatrogenic risk of miscarriage of 0.1% following invasive prenatal testing, all laboratories in France can carry out NIPT in VT pregnancies based on national recommendations [18]. NIPT in these pregnancies are reimbursed as twin pregnancies.

In this study, we describe our clinical experience using NIPT for common fetal trisomies in a large population of patients with a VT pregnancy. Confirmatory diagnostic testing was carried out on over 90% of the high-risk cases to determine concordance and the positive predictive value (PPV) in our study cohort. We found that NIPT is an effective prenatal screening approach for trisomy 21 in VT pregnancies but should be used carefully for trisomy 18 and 13.

## 2. Materials and Methods

Here, plasma samples from pregnant women with a gestational age of ≥10 weeks were collected and analyzed over a three-year period from January 2019 to December 2021 at a single laboratory in France. All patients with a VT were included in this retrospective observational study. We also included patients with an empty sac when the prescriber considered it as a vanishing twin, even if there was no proof of a prior cardiac activity. Prior to NIPT, all patients underwent an ultrasound scan that was normal except for the presence of a vanishing twin and one case with a nuchal translucency (NT) over 3.5 mm. Referral indications for patients were collected and included NIPT as personal choice (first-tier screen or MSS risk < 1/1000), first or second trimester MSS risk ≥ 1/1000, prior pregnancy with aneuploidy, and parental Robertsonian translocation. Informed consent was obtained from all subjects involved in the study. NIPT was carried out using the VeriSeq^TM^ NIPT Solution v2 assay as previously described [19,20]. Due to the presence of a vanishing twin, the analysis was carried out using the twin mode instead of the singleton mode. The VeriSeq NIPT assay uses a dynamic threshold metric (iFACT) that takes both fetal fraction (FF) and coverage information into account to determine whether or not a call can be made. This allows for accurate calls at low fetal fractions, and therefore the lab does not use a cut-off for low fetal fractions. Results were provided to the patients for common trisomies only (i.e., trisomy 21, 18, and 13). All patients with a high-risk result were advised to undergo an invasive diagnostic test for confirmation of the presence of the aneuploidy. Diagnostic confirmation was carried out using amniocentesis with karyotyping. Two comparison populations were also included in the study: the singleton pregnancies and the viable multiple gestation pregnancy samples collected over the time period of the study. Differences in FF and gestational ages between populations were assessed using *t*-tests. Differences in trisomy 21 prevalence between populations was assessed using a chi-square.

## 3. Results

### 3.1. Patient and Sample Details

Over a three-year period in our laboratory, 116,098 pregnant patients underwent NIPT: 105,560 singleton pregnancies, 9691 viable multiple gestation pregnancies (9479 twin gestations, 205 triplet gestations, and seven quadruplet gestations), and 847 (0.73%) vanishing twin pregnancies. The 847 vanishing twin pregnancies comprised the study population and included 694 cases with a VT and one surviving fetus, 148 cases with a VT and two surviving fetuses, one case with two vanishing twins and two surviving fetuses, and four cases with a VT and three surviving fetuses. The average gestational age (GA) for the study cohort was 15.6 weeks with a median of 14.5 weeks. Referral indications for our patient population are shown in Table 1. As can be seen, the majority (65%) of study patients underwent NIPT because of the multiple pregnancy (first-tier screen or MSS risk < 1/1000). Of the patients with a MSS risk ≥ 1/1000 (*n* = 284), the average risk was 1/451 (1/487 and 1/327 for first and second-trimester MSS risk, respectively) and the median risk was 1/450 (1/454 and 1/390 for first and second-trimester MSS risk, respectively); there was one case with an unknown risk value. All of the patients in the study cohort had an ultrasound examination. NT measurements were available for 786 (93%) patients in the study population. Of these, 764 (97%) cases had an NT value less than the 95th percentile. Of the remaining 22 cases, one (0.13%) had a NT value > 3.5 mm, which was a VT pregnancy with two surviving fetuses, and 21 (2.67%) had an NT between the 95th percentile and 3.5 mm including 17 cases with a VT and one surviving fetus and four cases with a VT and two surviving fetuses. No other anomalies were noted on ultrasound examination for any of the cases, except for the presence of a VT.

### 3.2. NIPT Results

There were a total of eight samples that did not obtain a result following the initial NIPT analysis, resulting in a first-pass no-call rate of 0.94%; all eight samples were resolved following a second blood draw including one that was positive for trisomy 21.

The average fetal fraction was 11.3%, the median was 10.0%, and the range was 2–29%. Fetal fraction results per group in the study cohort are shown in Table 2, along with FFs for the two comparison populations. We wanted to determinate whether the fetal fractions for the VT patient population were more comparable with the singleton pregnancy cohort or with the multiple pregnancies cohort. For VT pregnancies with one or two viable fetuses, the gestational age (GA) was slightly different to the comparator, but a one-week GA difference would likely have a minimal impact on FF. The FF for a VT pregnancy with one viable fetus was significantly higher than a singleton pregnancy (10.8% vs. 10.3%, *p* < 0.05) and significantly lower than a twin pregnancy with no VT (10.8% vs. 12.5%, *p* < 0.05). The FF for a VT pregnancy with two viable fetuses was found to be significantly higher than a twin pregnancy with no VT (*p* < 0.05), and not significantly different to a triplet pregnancy with no VT.

Based on our NIPT results, there was a total of 29 high-risk calls in our study cohort (3.42% vs. 0.69% in multiple pregnancy population): 14 for trisomy 21 (10 first-tier screening, three with MSS risk ≥ 1/1000, and one with a Robertsonian parental translocation rob (13; 21)), nine for trisomy 18 (six first-tier screening, three with MSS risk ≥ 1/1000), and six for trisomy 13 (five first-tier screening, one with MSS risk ≥ 1/1000) (Table 3).

The rate of trisomy 21 in our study cohort (1.65%; 14/847) was compared with the rate of trisomy 21 in the general multiple pregnancies population as the rate should be similar based on the number of fetuses (and then the theorical risk of a malsegregation leading to an aneuploidy) and the pipeline used. However, it was found to be significantly higher in the VT population than that seen in the general multiple pregnancies population (0.47%, 46/9691; *p* < 0.0001).

### 3.3. Outcome Information

All patients with a high-risk NIPT call but two underwent amniocentesis for confirmation of the aneuploidy in the surviving fetus/fetuses (see Table 4). Of the 14 trisomy 21 cases, one was lost to follow-up and one experienced an intrauterine fetal demise between the first trimester ultrasound and an ultrasound at 17.3 weeks (after the NIPT result). Of the remaining 12 trisomy 21 cases, there were six true positives and six false positives, giving a PPV of 50% (potential PPV range was 43–57% after accounting for the two cases without confirmation). Eight of the nine trisomy 18 cases and all six trisomy 13 cases were false positives. PPVs for the study cohort were compared with those observed in the viable multiple pregnancy population where fetal karyotype/array was available (Table 4). Although clinical outcomes were not specifically sought for pregnancies with a low risk NIPT, we expect that we would have been made aware of any false-negative result(s) because of the low number of cytogenetic French laboratories. No false-negative results were reported.

Finally, we looked at how the length of time between fetal demise diagnosis and sampling for NIPT affected the number of true-positive and false-positive cases in our cohort. The results were based on a total of 21 cases; we excluded two cases with an empty sac and six cases where the time from fetal demise to NIPT was unknown. As can be seen from Figure 1, no clear relationship was observed.

## 4. Discussion

Here, we describe our clinical experience with prenatal screening for common trisomies using NIPT in a large cohort of patients with a VT pregnancy. As can be seen from the results and unsurprisingly, there were many false-positive cases in our cohort. However, the PPV for trisomy 21 in the surviving fetus(es) was at least 50% (it could be 53% if we assumed that the intrauterine fetal death with a positive NIPT had a demise because of true fetal aneuploidy), which was still much higher than the PPV observed using traditional serum screening [21]. The PPVs observed in our study for trisomy 18 and 13 were much lower, however, there was only a small number of those cases in our cohort. The rate of no-call results was not increased in the study population, showing that VT is not a technical obstacle to performing NIPT.

When we looked at the rate of high-risk NIPT calls, we found a higher rate in our study cohort compared to the general multiple pregnancy population (3.42% vs. 0.69%) and a lower PPV in the viable fetus(es) (50%, 11%, and 0% for trisomy 21, 18, and 13, respectively). There are a couple of possible reasons for this.

The first is that in VT cases, cfDNA from the vanishing twin is still present in the maternal plasma and is the reason for the positive NIPT result. As outlined earlier, cfDNA from the demised twin has been reported to be present for up to 15 weeks post fetal demise [16]. As presence of a fetal aneuploidy in the VT may have been the cause for the fetal demise, this would result in a higher than expected rate of trisomy calls in our study population compared to the general multiple gestation population. Other studies have also noted that the presence of a VT can result in false-positive calls with NIPT [2,14]. A recent study suggested that patients with a VT pregnancy that either have a no-call or a high-risk result following NIPT should undergo a repeat NIPT after 15 weeks of gestation [22]. Here, we did not find a clear relationship between the number of weeks from fetal demise and the level of true-positives and false-positives in our study cohort. Based on this, it is not possible for us to suggest a specific length of time that patients should wait between the diagnosis of a vanished twin and undergoing NIPT.

Another possible reason for the higher level of positive results and lower PPV in our study cohort may be due to the fact that the NIPT assay was carried out using the twin mode, even if there was only one surviving fetus. As the number of fetuses in a VT pregnancy is not clearly defined, there is a chance that the performance of the assay for VT pregnancies with a single viable fetus at the time of NIPT would be better if run in the singleton mode. Indeed, we found that the FF for a VT pregnancy with one viable fetus was significantly higher than the FF in a singleton pregnancy, but significantly lower than the FF in a twin pregnancy with no VT. This is perhaps a future area of study that should be looked at to see if it can reduce the false-positive rate. For patients with a VT pregnancy and two surviving fetuses, the FF was not found to be significantly different to that of a triplet pregnancy with no VT and therefore the twin mode should not have any influence on the assay performance.

Even if performance of NIPT is lower than what we are used to in patient populations without VT, the 3.4% screen-positive rate (with no known false-negative results) supports that performing NIPT may be preferred over a systematic invasive procedure, as recommended by the American College of Obstetricians and Gynecologists and the Society of Maternal-Fetal Medicine. Indeed, the number of invasive tests generated by positive NIPT screening results, even if it is higher than in the patient population without VT, would remain much lower than systematic procedures in all VT pregnancies.

The PPV of 50% for trisomy 21 in the surviving fetus(es) in VT pregnancies reported here supports offering an invasive procedure to these patients. However, the low PPVs observed in our study for trisomies 18 and 13 in the surviving fetus(es) suggests that moving straight to an invasive test in these patients may not be the best recommended clinical management approach. Ultrasound monitoring could be offered along with a repeat NIPT several weeks later to see if the positive NIPT result resolves; a negative follow-up NIPT would suggest that the original NIPT positive result may have come from the cfDNA from the VT. If the ultrasound is normal and the repeat NIPT is negative, it would not be necessary to perform an invasive diagnostic test.

The main strength of this study is its originality, as few similar studies have been carried out to date. Nevertheless, the current study has several limitations. First, even though the cohort was large, the number of high-risk calls by NIPT was low, especially for T13 and T18. In addition, the consideration of an empty gestational sac without proof of the presence of a prior embryo as an vanishing twin pregnancy may have been a mistake as we cannot know if there is any contribution of genetic material from a gestational sac alone without confirmed presence of a prior embryo. Finally, it should also be noted that we do not have data on the outcomes of fetuses with negative NIPT. However, as above-mentioned, it is likely that we would have been informed of any false negatives in our study cohort. In conclusion, our study found that NIPT is an effective screening method for trisomy 21 in the surviving fetus(es) in VT pregnancies. Because of the iatrogenic risk of miscarriage, pregnant patients often prefer the noninvasive prenatal screening option over the systematic invasive procedure recommended by the American College of Obstetricians and Gynecologists and the Society of Maternal-Fetal Medicine. Nevertheless, patients should be counselled on the risk of a false-positive result when undergoing NIPT in a VT pregnancy, and that a high-risk call for trisomy 21 should be confirmed via amniocentesis before any decisions regarding the pregnancy are made. For VT pregnancies with a high-risk NIPT call for trisomy 13 or trisomy 18, an ultrasound and possibly a repeat NIPT later should be considered instead of amniocentesis to determine the presence of fetal anomalies due to the low PPV in the surviving fetus(es) for these trisomies.

## Figures and Tables

**Figure 1 genes-13-02027-f001:**
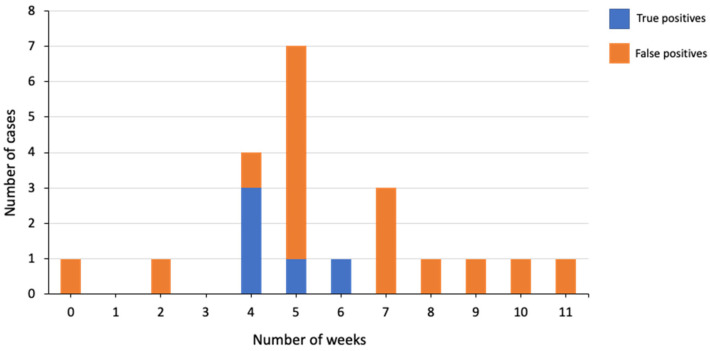
Relationship between the length of time from fetal demise diagnosis to NIPT and true-positive and false-positive calls.

**Table 1 genes-13-02027-t001:** Referral indications for study patients (*n* = 847).

Referral Indication	VT + 1 Surviving Fetus	VT + 2 Surviving Fetuses	VT + 3 Surviving Fetuses	Total
Parental Robertsonian translocation	2	0	0	2
First-trimester MSS ≥ 1/1000	183	1	0	184
Second-trimester MSS ≥ 1/1000	99	1	0	100
Prior pregnancy with aneuploidy	9	0	0	9
First-tier screening	386	147	4	537
First-trimester MSS < 1/1000	12	0	0	12
Second-trimester MSS < 1/1000	3	0	0	3
Total	694 (81.9%)	149 (17.6%)	4 (0.5%)	847

MSS, maternal serum screening; VT, vanishing twin.

**Table 2 genes-13-02027-t002:** Fetal fractions for the study cohort and comparison cohorts.

Cohort	Mean FF, %	Median FF, %	Mean GA, wk	Median GA, wk
**Study cohort**				
VT + 1 surviving fetus	10.8	10	15.6	14.4
VT + 2 surviving fetuses	13.6	13	15.8	15
VT + 3 surviving fetuses	18	17.5	13.4	13.3
Total	11.3	10	15.6	14.5
**Multiple pregnancies with no VT ***				
2 fetuses	12.5	12	14.8	13.5
3 or 4 fetuses	14.4	14	14	13
Total	12.5	12	14.8	13.5
**Singleton pregnancies ^‡^**	10.3	10	16.7	15.4

FF, fetal fraction; GA, gestational age; VT, vanishing twin. * 14 cases were excluded from the FF calculations due to unknown FF (they were no-call samples). ^‡^ Nine cases were excluded due to having a GA less than 10 weeks or an unknown FF.

**Table 3 genes-13-02027-t003:** NIPT results in our study cohort.

NIPT Result	VT + 1 Surviving Fetus	VT + 2 Surviving Fetuses	VT + 3Surviving Fetuses	Total, *n* (%)	Multiple Pregnancies with No VT, *n* (%)
Trisomy 21	13 *	1	0	14 (1.65)	46 (0.47)
Trisomy 18	9	0	0	9 (1.06)	11 (0.11)
Trisomy 13	6	0	0	6 (0.71)	10 (0.10)
Low risk	666	148	4	818 (96.58)	9610 (99.16)
No call	0	0	0	0	14 (0.14)

VT, vanishing twin. * One case had a dup (21) (q21.1q22.2).

**Table 4 genes-13-02027-t004:** Outcomes for high-risk NIPT cases.

Trisomy (*n*)	True Positive	False Positive	PPV	PPV for Multiple Pregnancies with No VT
Trisomy 21 (14 *)	6	6	50%	79.5% (35/44)
Trisomy 18 (9)	1	8 ^‡^	11.1%	50% (5/10)
Trisomy 13 (6)	0	6	0%	0% (0/7)

PPV, positive predictive value. * One case was lost to follow-up and one case had an intrauterine fetal demise. ^‡^ One case was positive for 47, XXY.

## Data Availability

Deidentified data that support the findings of this study are available upon reasonable request from the corresponding author within 12 months of publication. The clinical outcome data in this study were obtained from patient records and therefore will not be made available through a database due to privacy concerns. Proposals must include a detailed and sound methodological approach including a statistical analysis plan, as would be reasonably required for the purposes of publication in a peer-reviewed scientific journal.

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
