# Peer review of "Noninvasive Prenatal Screening for Trisomy 21 in Patients with a Vanishing Twin"

_genes, 2022, doi:10.3390/genes13112027_

Round 1

Reviewer 1 Report

The article titled ‘Noninvasive prenatal screening for trisomy 21 in patients with a vanishing twin’ overall represents very important clinical data studies of French population related to vanishing twins VT. This imposes a real time impact of prenatal screening for fetal aneuploidy. However some minor errors have been observed. Which are listed below:

1. In introduction abbreviations used should be coded first and then used in the text (for example IVF)

2. Isn’t it number of VT patients 847 is high for the period of three year from a single population? Are there any ethical or demographical association studies available for such anomalies?

3. Which of the two comparative populations were selected for studies and on which preference?

4. False positive outcomes observed exceed the true results. Any possible reason?

5. is there any risk of fetal demise associated with prenatal testing and the gestational week for test?

6. What could be the possible molecular or genetic reason of trisomies leading to VT?

7. Can you please elaborate why one should prefer NIPT over IPT? It would be worth sharing for the readers.

Author Response

Thank you very much for your comments. Please find our responses below (the new version of the revised paper is uploaded) :

1- In introduction abbreviations used should be coded first and then used in the text (for example IVF)

Response: Thank you for your comment. The abbreviations for IVF, NT, FF, and IUFD have been spelled out upon first use in the text (see lines 33, 56, 66, and 74).

  1. Isn’t it number of VT patients 847 is high for the period of three year from a single population? Are there any ethical or demographical association studies available for such anomalies?

Response: The prevalence of VT in our population is 0.73%. No conclusion could be drawn on that prevalence because 1) the prevalence of VT in population with a spontaneous conception is not clear and 2) we don’t know in our population the % of IVF pregnancies which are known to have a prevalence of VT of 10% to 39%.

  1. Which of the two comparative populations were selected for studies and on which preference?

Response: In this study we used two comparison populations, namely the singleton pregnancies population and the viable multiple gestation pregnancy population. Both comparison populations were used to compare the FF in an effect to see if the VT population was more similar to the singleton pregnancies population or the population with viable multiple gestations. The results of these comparisons is noted on lines 121–128 and Table 2. In addition, the rate of positive results and the PPV of the VT study cohort were compared with those of the multiple gestations population because the risk should be comparable regarding the number of fetuses and the analysis pipeline. We have provided an explanation for our choice of comparison populations on lines 134-137. The result of the comparison regarding the rate of positive result is lines 134-137, 187-200 and Table 3. The result of the comparison regarding the PPV is shown in Table 4.

  1. False positive outcomes observed exceed the true results. Any possible reason?

Response: We have provided two reasons for this in the manuscript. The first is that the viable fetus(es) don’t have aneuploidy but aneuploidy was present in the vanished twin which would explain the fetal demise. The cfDNA from the affected vanished twin could still be present in the maternal circulation, leading to the discordant result. The second explanation that we provided is that the NIPT assay was carried out using the twin mode, and it is possible that the pipeline for two fetuses does not perform as well in a VT population. This is because we don’t know, when the test is being performed, if there are one or two placentas that are releasing cfDNA. Our explanations are provided on lines 201-225.

  1. is there any risk of fetal demise associated with prenatal testing and the gestational week for test?

Response: We have added information regarding the iatrogenic risk of invasive prenatal testing to the manuscript (see lines 49-50).

  1. What could be the possible molecular or genetic reason of trisomies leading to VT?

Response: The link between chromosomal anomalies and miscarriages has been added (see lines 43-44).

  1. Can you please elaborate why one should prefer NIPT over IPT? It would be worth sharing for the readers.

Response: This has been added in the conclusion lines 252-255.

  1. In introduction abbreviations used should be coded first and then used in the text (for example IVF)

Response: Thank you for your comment. The abbreviations for IVF, NT, FF, and IUFD have been spelled out upon first use in the text (see lines 33, 56, 66, and 74).

  1. Isn’t it number of VT patients 847 is high for the period of three year from a single population? Are there any ethical or demographical association studies available for such anomalies?

Response: The prevalence of VT in our population is 0.73%. No conclusion could be drawn on that prevalence because 1) the prevalence of VT in population with a spontaneous conception is not clear and 2) we don’t know in our population the % of IVF pregnancies which are known to have a prevalence of VT of 10% to 39%.

  1. Which of the two comparative populations were selected for studies and on which preference?

Response: In this study we used two comparison populations, namely the singleton pregnancies population and the viable multiple gestation pregnancy population. Both comparison populations were used to compare the FF in an effect to see if the VT population was more similar to the singleton pregnancies population or the population with viable multiple gestations. The results of these comparisons is noted on lines 121–128 and Table 2. In addition, the rate of positive results and the PPV of the VT study cohort were compared with those of the multiple gestations population because the risk should be comparable regarding the number of fetuses and the analysis pipeline. We have provided an explanation for our choice of comparison populations on lines 134-137. The result of the comparison regarding the rate of positive result is lines 134-137, 187-200 and Table 3. The result of the comparison regarding the PPV is shown in Table 4.

  1. False positive outcomes observed exceed the true results. Any possible reason?

Response: We have provided two reasons for this in the manuscript. The first is that the viable fetus(es) don’t have aneuploidy but aneuploidy was present in the vanished twin which would explain the fetal demise. The cfDNA from the affected vanished twin could still be present in the maternal circulation, leading to the discordant result. The second explanation that we provided is that the NIPT assay was carried out using the twin mode, and it is possible that the pipeline for two fetuses does not perform as well in a VT population. This is because we don’t know, when the test is being performed, if there are one or two placentas that are releasing cfDNA. Our explanations are provided on lines 201-225.

  1. is there any risk of fetal demise associated with prenatal testing and the gestational week for test?

Response: We have added information regarding the iatrogenic risk of invasive prenatal testing to the manuscript (see lines 49-50).

  1. What could be the possible molecular or genetic reason of trisomies leading to VT?

Response: The link between chromosomal anomalies and miscarriages has been added (see lines 43-44).

  1. Can you please elaborate why one should prefer NIPT over IPT? It would be worth sharing for the readers.

Response: This has been added in the conclusion lines 252-255.

Reviewer 2 Report

The manuscript of Kleinfinger and colleagues entitled "Non-invasive prenatal screening for trisomy 21 in patients with a vanishing twin" studies the feasibility and outcomes of performing NIPS in pregnancies affected by a vanishing twin or VT.  

Overall, the article is well written and is careful in reporting its conclusions based on the available data. There are several important points raised by this article that are valuable for providers who encounter NIPS results. The first is that at least for this Veriseq platform, there does not appear to be a technical obstacle to performing NIPS in such cases. Second, that there was an elevated number of T21 cases found among the VT pregnancies with accompanying elevations in fetal fraction. I appreciated that ultrasounds, NTs and follow up diagnostic testing by amniocentesis were performed in the majority of subjects which allowed examination of the PPV. Importantly, this showed that PPV was decreased (50% vs 79.5%) when compared to twin/multiple pregnancies without a VT.  Further, the PPV was poor for T18 and T13 with a high FP rate and the authors appropriately cautioned that for those trisomies, actual diagnostic testing should be pursued. 

The article adds to the literature surrounding this rapidly evolving technology. I had a few questions which are minor points to consider:

1. Is it known if any or how many of the pregnancies resulted from IVF and if any PGT had been performed?

2. In Table 1 where the indication was parental balanced Robertsonian translocation or prior aneuploidy, were any of these cases among the positive screens or positive results?

3. The Veriseq platform appears to be a WGS-based platform. Do the authors have any data on whether other assays perform similarly? Are there technical advantages or disadvantages to consider based on the use of VeriSeq versus other technology that is widely used?

4. Can the authors comment if there was only one laboratory performing the assays from multiple referring centers? If not, did they ensure that everyone is using the same assay kit and metrics?

5. Is the elevated percentage of T21 found in VT pregnancies associated with a maternal age effect in those pregnancies? If not, can the authors more clearly state what they believe may be the mechanism for this finding?

6. What is the range of fetal fraction in the VT and what is the lab cut-off for low fetal fraction (methods)?

Author Response

Thank you very much for your comments. Please find our responses below (the new version of the revised paper is uploaded) 

  1. Is it known if any or how many of the pregnancies resulted from IVF and if any PGT had been performed?

Response: Unfortunately, we don’t have this information.

  1. In Table 1 where the indication was parental balanced Robertsonian translocation or prior aneuploidy, were any of these cases among the positive screens or positive results?

Response: We have added this information to the manuscript (see lines 129-133).

  1. The Veriseq platform appears to be a WGS-based platform. Do the authors have any data on whether other assays perform similarly? Are there technical advantages or disadvantages to consider based on the use of VeriSeq versus other technology that is widely used?

Response:  A 2015 publication by Curnow et al. ( DOI: 10.1016/j.ajog.2014.10.012) looked at the detection of vanishing twin pregnancies (along with triploid and molar pregnancies) by a single-nucleotide polymorphism-based NIPT. This study found that the NIPT assay was able to successfully identify VT, and that VT pregnancies had a significantly higher median maternal age than twin cases with no significant difference in the average fetal fraction of VT and twin pregnancies. Conversely in our study we found that the FF for a VT pregnancy with 1 viable fetus was significantly lower than a twin pregnancy with no VT (10.8% versus 12.5%, p<0.05) and the FF for a VT pregnancy with 2 viable fetuses was found to be significantly higher than a twin pregnancy with no VT (p<0.05).

A technical advantage of the VeriSeq assay is that is uses a dynamic threshold metric (iFACT) which takes both fetal fraction and coverage information into account to determine whether or not a call can be made, allowing accurate calls at low fetal fractions (outlined in response to Question 6 below).

  1. Can the authors comment if there was only one laboratory performing the assays from multiple referring centers? If not, did they ensure that everyone is using the same assay kit and metrics?

Response:   All of the study including sample collection and analysis was carried out at a single laboratory as noted on line 62 of the manuscript.

  1. Is the elevated percentage of T21 found in VT pregnancies associated with a maternal age effect in those pregnancies? If not, can the authors more clearly state what they believe may be the mechanism for this finding?

Response: As noted in response to Reviewer 1 above (Q4), we have provided two reasons for this in the manuscript. The first is that the viable fetus(es) don’t have aneuploidy but aneuploidy was present in the vanished twin which would explain the fetal demise. The cfDNA from the affected vanished twin could still be present in the maternal circulation, leading to the discordant result. The second explanation that we provided is that the NIPT assay was carried out using the twin mode, and it is possible that the pipeline for two fetuses does not perform as well in a VT population. This is because we don’t know, when the test is being performed, if there are one or two placentas that are releasing cfDNA. Our explanations are provided on lines 201-225.

  1. What is the range of fetal fraction in the VT and what is the lab cut-off for low fetal fraction (methods)?

Response: The range for fetal fraction in the VT study cohort has been added (lines 117-118). The VeriSeq NIPT assay uses a dynamic threshold metric (iFACT) which takes both fetal fraction and coverage information into account to determine whether or not a call can be made. This allows accurate calls at low fetal fractions, and therefore the lab does not use a cut-off for low fetal fractions. 
